# Alternative Promoter Use Governs the Expression of IgLON Cell Adhesion Molecules in Histogenetic Fields of the Embryonic Mouse Brain

**DOI:** 10.3390/ijms22136955

**Published:** 2021-06-28

**Authors:** Toomas Jagomäe, Katyayani Singh, Mari-Anne Philips, Mohan Jayaram, Kadri Seppa, Triin Tekko, Scott F. Gilbert, Eero Vasar, Kersti Lilleväli

**Affiliations:** 1Department of Physiology, Institute of Biomedicine and Translational Medicine, University of Tartu, 19 Ravila Street, 50411 Tartu, Estonia; toomas.jagomae@gmail.com (T.J.); marianne.philips@ut.ee (M.-A.P.); jmohan80@gmail.com (M.J.); kadriseppa@gmail.com (K.S.); eero.vasar@ut.ee (E.V.); kersti.lillevali@ut.ee (K.L.); 2Centre of Excellence in Genomics and Translational Medicine, University of Tartu, 50090 Tartu, Estonia; 3Laboratory Animal Centre, Institute of Biomedicine and Translational Medicine, University of Tartu, 14B Ravila Street, 50411 Tartu, Estonia; 4The Instituto Gulbenkian de Ciência, Rua da Quinta Grande 6, 2780-156 Oeiras, Portugal; ttekko@igc.gulbenkian.pt; 5Department of Biology, Swarthmore College, Swarthmore, PA 19081, USA; sgilber1@swarthmore.edu

**Keywords:** IgLON, *Lsamp*, *Ntm*, *Opcml*, *Negr1*, alternative promoter, cell adhesion molecules, embryonic mouse brain, pallium

## Abstract

The members of the IgLON superfamily of cell adhesion molecules facilitate fundamental cellular communication during brain development, maintain functional brain circuitry, and are associated with several neuropsychiatric disorders such as depression, autism, schizophrenia, and intellectual disabilities. Usage of alternative promoter-specific *1a* and *1b* mRNA isoforms in *Lsamp*, *Opcml*, *Ntm*, and the single promoter of *Negr1* in the mouse and human brain has been previously described. To determine the precise spatiotemporal expression dynamics of *Lsamp*, *Opcml*, *Ntm* isoforms, and *Negr1*, in the developing brain, we generated isoform-specific RNA probes and carried out in situ hybridization in the developing (embryonic, E10.5, E11.5, 13.5, 17; postnatal, P0) and adult mouse brains. We show that promoter-specific expression of IgLONs is established early during pallial development (at E10.5), where it remains throughout its differentiation through adulthood. In the diencephalon, midbrain, and hindbrain, strong expression patterns are initiated a few days later and begin fading after birth, being only faintly expressed during adulthood. Thus, the expression of specific IgLONs in the developing brain may provide the means for regionally specific functionality as well as for specific regional vulnerabilities. The current study will therefore improve the understanding of how IgLON genes are implicated in the development of neuropsychiatric disorders.

## 1. Introduction

The first sign of vertebrate brain development is the appearance of the neural plate, which gives rise to neuroepithelial cells (NECs), the precursors of neural progenitor cells (NPCs). Early histogenetic processes of neural tissue formation encompasses massive proliferation and migration of progenitor cells, followed by neuronal differentiation together with neuritogenesis and axon guidance to appropriate targets for creating synaptic connections [1,2,3]. These processes are highly coordinated by spatiotemporal crosstalk of signaling cues and gene regulatory pathways, accounting for proteome diversifications by generating multiple alternative mRNA isoforms [4,5,6,7]. Cell adhesion molecules function as key elements for all these developmental milestones.

Members of neuronal cell adhesion molecules of the IgLON superfamily, LSAMP, (limbic system associated membrane proteins), NTM (neurotrimin), OPCML (opioid-binding cell adhesion molecule), harbor two alternative promoters (*1a* and *1b*), leading to transcripts that encode proteins with alternative N-terminal sequences. Other members, NEGR1 and IgLON5 have instead a single promoter [8,9]. The IgLON superfamily of brain glycoproteins carry three Ig domains and are anchored to neural and oligodendrocyte cell membranes by glycosylphosphatidylinositol (GPI) [10,11]. Homophilic and heterophilic intra-family interactions of IgLONs on the plane of cell membranes can modify the context-dependent functional aspects of neural development, maintenance, and plasticity [12,13,14,15,16,17,18,19,20,21,22]. In humans, several polymorphisms at IgLON loci and imbalances in IgLON expression levels are associated with cognitive ability andwide variety of neuropsychiatric disorders, such as schizophrenia, major depression, bipolar disorder, and autism [23,24,25,26,27,28]. Specific phenotypes appear to be influenced by all IgLON genes. For example, autism has been shown to be linked with the loci of *NEGR1* [29] *NTM/OPCML* [30] and *LSAMP* [31]. Similarly, differences in cognitive ability have been associated with *NEGR1* [32] and the loci of all IgLONs significantly associate with the educational attainment [33].

Genetic deficiencies of *Lsamp*, *Ntm*, *Negr1* in mice result in early developmental anomalies due to defects of corticogenesis, neuritogenesis, axon guidance, myelinization, as well as adult abnormalities of neurogenesis and maintenance [11,17,20,21,22,34]. *Lsamp*-deficient mice have alterations in neurotransmitter regulation, including the increased activity of the serotonergic system, imbalanced GABA_A_ receptors activity, as well as decreased sensitivity towards amphetamine, a condition that is also found in other IgLON deficient mice (*Ntm* and *Lsamp/Ntm* double-mutant mice) [20,35,36,37,38,39,40]. Behavioral studies performed with *Lsamp*, *Ntm*, *Negr1* and double *Lsamp/Ntm* deficient mice showed impairment of locomotor, cognition, mood and social behavior [37,38,39,40,41,42]. Moreover, *Negr1* deficiency in mice produces significant volumetric changes in the brain morphology, including decreased brain size, ventricular enlargement and the depletion of hippocampal parvalbumin (PV)^+^ neurons [34,42]. The above characteristics of IgLON deficiencies in mice are very similar to those of people with certain neuropsychiatric patients and can serve as model endophenotypes for these conditions.

Our qPCR-based studies on the adult mouse brain have demonstrated differential and largely complementary expression profiles driven by alternative promoters of *Lsamp*, *Ntm*, and *Opcml* genes [1,4,9]. So far, promoter-specific neuroanatomical expression has been characterized only for *Lsamp*, showing that alternative promoters (*1a* and *1b*) are employed differentially, with few overlapping areas from embryonic age E12.5 and that they help regulate emotional and social behaviors. The *1a* isoform is abundant in the classic limbic structures such as the hippocampus, amygdala, and the caudal subgroup of raphe nuclei, whereas the *1b* isoform is expressed in the thalamic sensory nuclei, isocortical sensory areas, and in medial raphe nuclei [39,41].Alternative promoter-specificIgLONs expression profiles were also studied in the dorsolateral prefrontal cortex (DLPFC) of patients with schizophrenia in the autopsy samples, and altered expression levels from *Ntm 1a*, *1b* isoforms and *Negr1* have been reported [28].

Despite the potentially important role of IgLONs in establishing functional neural circuits, little is known about the spatial and temporal expression of alternative promoter-specific patterns in the embryonic and postnatal brain. Therefore, to investigate the spatial expression from *Negr1*, as well as from the two alternative promoters of the *Lsamp*, *Ntm*, and *Opcml*, isoform-specific probes for in situ hybridization were used to label the IgLON mRNA expression patterns in developing mouse brain. Our data show an early onset of IgLON expression at E10.5, the time when active neuroepithelial expansion, neurogenesis, and neuronal migration occur. Each member of the IgLON superfamily appears in the embryonic brain in a characteristic dynamic pattern that closely approximates its adult distribution in the central nervous system (CNS).

## 2. Results

### 2.1. Initiation of the Expression of IgLON Members in the Early Embryonic Mouse Brain (E10–13)

The first sign of the expression of *Lsamp*, *Negr1*, *Ntm*, and *Opcml* in the CNS coincided in mid gestational stages, at the time when active neurogenesis and neuronal migration occurs. Prior to the neuroepithelium entering its massive neurogenesis phase at E10.5, expression is already observable for *Lsamp 1a*, *Negr1*, *Ntm 1a*, and *Opcml 1b* in the neural tube, ventricular zone, and around the ventricular vesicles (Appendix A). Whole-mount insituhybridization of E11.5 embryos shows persistent expression of IgLONs at E11.5 in the dorsal part of developing brain vesicles (Figure 1A–N). Intense signal is observable in telencephalic vesicles (Figure 1A–G) and at the level of pontine flexure (Figure 1A–H). Segmental expression of *Lsamp 1a* (Figure 1H), *Ntm 1a*, *Ntm 1b*, *Opcml 1a* and *Opcml 1b* (Figure 1K–N) lies bilaterally along developing neural tube, whereas the expression of *Lsampuni* (Figure 1I), *Negr1* (Figure 1J), and *Opcml 1a* (Figure 1M) is observable at the level of the limbs. Expression of IgLONs from alternative promoters lies ventrally in the pharyngeal pouches fading in the dorsal direction. From all observed probes, *Lsamp*, *Ntm* and *Negr1* gave positive signal in the developing inner ear (Figure 1A–E). Faint signal is seen in presumptive trigeminal ganglion with *Lsamp 1a/uni*, *Ntm 1a* and *Opcml 1b* probes (Figure 1A,B,E,G).

#### 2.1.1. Forebrain (E13.5)

At E13.5 the presence of all IgLON transcripts is observed in the dorsal pallium in a ventricular-subventricular descending gradient. The expression patterns of *Lsamp 1a/uni* are overlapping and detected in the ventricular/subventricular zone of dorsal, ventral and lateral pallium (Figure 2A,B,D,E). A faint signal is seen as a column extending from the corner of the lateral ventricle to the pial side of the presumptive insular/piriform cortices and pallial amygdala (Figure 2B,E and Figure 3A,B). As with *Lsamp 1a* and *uni* probes, the *Ntm 1a/1b* probes are detected in the ventricular and subventricular zone (SVZ) of dorsal, ventral and lateral pallium (Figure 2G,H,J,K). However, moving from the dorsal pallium towards the lateral pallium, the expression pattern splits distinctively into two separate/different domains, forming a crescent shape in the SVZ of the putative lateral and ventral pallium (Figure 2H,K). A similar, but fainter, column of expression is also observed from the ventricular zone of lateral and ventral pallium to the pial side (Figure 2H,K and Figure 3C,D).

A strong signal from *Ntm 1a/1b* probes is seen in the pial side of the developing insular/piriform cortices (Figure 2H,K), and signals from *Opcml 1a/1b* probes are observable in the ventricular/subventricular zone of the pallium (Figure 2M,N,P,Q and Figure 3E,F). A similar expression pattern continues in the lateral ganglionic eminence (Figure 2N,Q). Here, a uniform *Opcml 1a/1b* signal is restricted to the lateral ganglionic eminence (Figure 2N,Q), while columns of faint expression of *Opcml 1a/1b* bordering the lateral ganglionic eminence appear migrating ventro-laterally towards the pial side (Figure 2M,N).

In the pial side of the developing lateral and ventral pallium, the signal of *Opcml 1b* becomes intense (Figure 2Q and Figure 3F). *Negr1* is expressed in the ventricular/subventricular zone of dorsal pallium (Figure 2S,T and Figure 3G). On the pial side of ventral and lateral pallium the expression of *Negr1* appears especially strong and opposes the ventricular/subventricular zone by forming a crescent in the pallium/subpallium boundary. Ventrally, *Negr1* expression continuously lines the pial side of the allocortex (Figure 2T).

#### 2.1.2. Midbrain and Spinal Cord (E13.5)

In the developing midbrain, faint-to-medium level expression patterns of all the IgLON transcripts are detectable in the dorso-rostral part of neuroepithelium surrounding the 3rd ventricle (Figure 2A,D,G,J,M,P,S). All IgLON transcripts appear in the developing spinal cord (sc) in distinct regions: *Lsamp 1a/uni* expression is seen dorso-laterally in the developing sensory input area of sc, and the roof plate is marked strongly by *uni* (Figure 2C,F); *Ntm 1a* and *1b* occupy dorsal the portion of neural tube in a complementary manner: *Ntm 1a* is strongly expressed immediately below the laterally positioned *Ntm 1b*-positive domain (Figure 2I,L). Different isoform expression patterns are also detectable with *Opcml*: while uniform *Opcml 1a* signal is detectable throughout the developing sc, being somewhat stronger in the motor region, the *Opcml 1b* transcript occupies the lateral portion of sc (Figure 2O,R). *Negr1* is strongly expressed in the dorsolateral domain of developing sc, being complementary to *Lsamp 1a/uni* with the exception seen at the roof plate (Figure 2U,C,F). In addition to their neural expression, there are also clear signals of *Lsamp 1a/uni*, *Ntm 1a*, *Opcml 1a* and *Negr1* in above and dorsolateral spinal cord (Figure 2C,F,I,L,O,U).

### 2.2. The Expression Dynamics of IgLON Members in the Perinatal Stages (E17, P0)

Expression of IgLON members is established in most primordial brain structures at E17, coinciding with formation and stabilization of neural circuits. Signals from IgLON probes form distinct but overlapping domains within developing structures. Here, the expression patterns from alternative promoters of *Lsamp*, *Ntm*, and *Opcml* are principally complementary. As the brain matures, the IgLON expression patterns are maintained in their original locations. 

#### 2.2.1. Cerebral Cortex and Hippocampus

At E17, neurogenesis of the outermost layers of cortical plate is just ending and neuronal morphogenesis and migration occurs.At this critical developmental stage, all IgLON transcripts are abundantly present in the forming neocortex. Presence of IgLON transcripts is summarized in Table 1.

Markedly, the expression of *Lsamp 1a* shows labeling below the marginal zone of insula and neocortex, where it delineates the subplate region (future layer Vib) (Figure 4A–C). The expression of *Lsamp 1a* is observed in the subventricular zone of the neocortex (Figure 4A,B), while *Lsamp 1b* expression pattern is localized into the cortical plate of presumptive barrel cortex (Figure 4D–F) and shows labeling of both the insula and neocortex, but not claustrum. At P0, the expression of *Lsamp 1a* is mainly restricted to layer 1 of the neocortex, except at the medial cingulate cortex, where labeling extends weakly from subiculum to CA1 (Figure 5A–C). Localization pattern of *Lsamp 1b* is extended dorsally and additionally to the upper layers of the neocortex.

At E17, *Ntm* transcripts are expressed continuously from Pir to the developing hippocampus, where the expression domain is narrower in width and exhibits an outside-inside strength gradient (Figure 4G–L, Appendix A). This signal, however, is absent from the marginal zone. Expression of *Ntm 1a* shows superficial insular labeling (Figure 4G). Meanwhile cortical labeling stops medially at the parahippocampal subiculum (Figure 4G–I). The *Ntm 1b* transcript is more enriched laterally, whereas only a weak signal is detectable in the developing hippocampus (Figure 4J–L, Appendix A). Expression of *Ntm 1a* at P0, shows a relatively strong signal throughout the developing sensory and motor cortices (Figure 5G–H), where the continuous expression from the subiculum to the hippocampus weakens and disappears in the presumptive CA3 region (Figure 5H,I). The pattern of *Ntm 1b* is similar to *1a* transcript at P0 (Figure 5K,L).

At E17.5 *Opcml 1a* is detected throughout the developing cortex, and the signal extends from Pir continuously to subiculum and CA regions of the developing hippocampus (Figure 4M,N, Appendix A). The *Opmcl 1b* signal extends throughout the entire width of cortical plate, being stronger laterally (Figure 4P–R). *Opcml 1b* extends into CA fields, but not into the DG (Figure 4Q,R, Appendix A). In the newborn (P0) mouse cortex, the expression of *Opcml 1a* resembles the expression at E17, although medio-lateral and dorso-ventral gradients are appearing (Figure 5M–O). Expression of *Opcml 1b* in the developing cortex is shifted towards upper layers (Figure 5P–R) and is clearly defined in layer 6b (Figure 5Q).

*Negr1* expression in the cortical plate declines from anterior to posterior and lateral to medial. *Negr1* is seen in the subventricular zone of dorsal pallium. We also observe strong expression in the layer 6b continuous with the claustrum/insula (Figure 4S–U). Distinct labeling is also seen in the dorsal endopiriform nucleus, deep to the presumptive olfactory cortex. Expression of *Negr1* at P0 remains relatively similar to embryonic stages, although labeling in the first layer has disappeared.

#### 2.2.2. Subcortical Structures and Diencephalon

*Lsamp 1a* at E17 labels the transition between CPu and ventral striatum continuously with the lateral and dorsal part of the striatum (Figure 4A). Expression in the thalamus is mainly periventricular (Figure 4B,C) with some indication of pre-thalamic positive elements above the peduncle (Figure 4C, Appendix A). Expression of *Lsamp 1a* is also seen at the ventromedial hypothalamic nucleus (VMH) (Figure 4C). *Lsamp 1b* expression emerges from the ventricular zone of subpallium extending caudally around the third ventricle (Figure 4D–F). Dense labeling of the dorsal endopiriform nucleus is also seen (Figure 4D). At P0 *Lsamp 1a* seems restricted to intralaminar thalamic elements and to the parafascicular nucleus avoiding the retroflex tract (Figure 5A–C). At both perinatal stages, the expression of *Lsamp 1a* appears in the striatum, pallial- and medial amygdala (Figure 4B,C and Figure 5B). *Lsamp 1b* expression at P0 is maintained as in E17 (Figure 5D–F).

At E17 *Ntm 1a* expression is seen as a latero-medial gradient in the lateral part of CPu (Figure 4G). Extensive labeling is observable in the thalamus, except for the lateral geniculate nucleus at the surface (Figure 4H,I). Expression is also seen in prethalamus elements and intergeniculate nucleus (Figure 4H,I). Although *Ntm 1a* is absent from amygdala (Figure 4G–I) the signal of *Ntm 1b* is detectable in the pallial and medial amygdala at E17 (Figure 4K,L) and P0 (Figure 5K,L). At E17 *Ntm 1b* expression in the CPu is complementary to *Ntm 1a*, absent laterally (Figure 4J). *Ntm 1b* shows a similar prethalamic expression pattern, although the superficial pre-geniculate nucleus is positive (Figure 4K–L, Appendix A). At P0 *Ntm 1b* expression in the CPu forms a lateral to medial gradient (Figure 5G,J).

*Opcml 1a* at E17 is expressed primarily in the superficial thalamus as well as in the medial Hb (Figure 4N, Appendix A). Staining is seen at VMH nucleus and superficially in the optic tract (Figure 4O). Expression of *Opcml 1b* shows subpial labeling along the optic tract and the presumptive arcuate nucleus (Figure 4Q,R). Expression of *Opcml 1b* invades medially to the olfactory tuberculum (Figure 4Q). Expression of *Opcml 1b* is prominent in the reticular-, ventromedial- and lateral thalamic nuclei at E17 (Figure 4Q,R) and remains in the reticular- and ventromedial thalamic nuclei at P0 (Figure 5Q,R, Appendix A). Strong expression of *Opcml 1b* is seen in the peduncle and thalamic nuclei avoiding the domains positive for *Opcml 1a* (Figure 4P–R). Expression of *Opcml 1a/1b* in the CPu resembles that of *Ntm 1a/1b* (Figure 4M,P). The expression of *Opcml 1a* at E17 is localized diffusely in the presumptive amygdala region (Figure 4N,O, Appendix A), whereas at P0 is expressed in the basolateral and lateral amygdala (Figure 5N,O). *Opcml 1b* maintains broad expression in the pallial amygdala at E17 (Figure 4R) and becomes concentrated to specific nuclei at P0 (Figure 5P,Q).

*Negr1* expression at E17 ascends above the lateral angle of the lateral ventricle, becoming strongest in the lateral septum (Figure 4S,T). Expression in the mantle of the ventral pallium is limited medially by the pallial-subpallial boundary and extends into the pallial amygdala (Figure 4T). At P0, the expression of *Negr1* covers the cortical, basolateral, medial, and basomedial amygdala (Figure 5S–U). At E17, the prethalamic subgeniculate and zona incerta formations are distinctly positive just above the peduncle leaving the pregeniculate nucleus negative (Figure 4T,U). Strong expression is observable at the dorsal, medial and basolateral thalamic complexes (Figure 4T,U and Figure 5T,U).

### 2.3. The Expression of IgLON Members in the Adult Mouse Brain

#### 2.3.1. Cerebral Cortex and Hippocampus

The adult brain expression pattern of *Lsamp1a*, *1b* and *uni* is described previously by [39]. In the adult brain, the expression of *Ntm 1a* is observed throughout the cortical layers, being stronger in upper borders of 2/3 and IV (Figure 6A–C). At the level of septum and anterior commissure, expression of *Ntm 1a* is evident in the barrel and motor cortices, whereas faint expression is seen in the sensory region of cortex (Figure 6A). In contrast to *Ntm 1a*, expression of *Ntm 1b* is clearly visible in the barrel cortex, with weak or no expression in the sensory and motor regions (Figure 6D). The expression of *Ntm 1b* is enriched in upper layers of barrel, visual and auditory regions (Figure 6E,F). *Ntm 1a* displays uniform expression in piriform cortex, Ectorhinal and Entorhinal (Figure 6A–C) whereas strong expression of *Ntm 1b* is seen in Pir and Ent (Figure 6D–F). In the hippocampus, *Ntm 1a* expression is enriched in CA1 while a relatively weak signal is observable in CA2, CA3 and DG (Figure 6B,C). A strong signal of *Ntm 1b* is evident rostrally in CA and DG and fragments in caudal direction (Figure 6E,F).

*Opcml 1a* is observable throughout cortical layers 2–6, being intense at borders (Figure 6G–I), while *Opcml 1b* is restricted to the sensory cortex and forms a gradient inverse to *Opcml 1a*, being stronger in the middle layers (Figure 6J–L). Although *Opcml 1a* is weak in Pir, Ect and Ent (Figure 6G–I), intense staining of *Opcml 1b* in these areas was seen (Figure 6J–L). The expression of *Opcml 1a* is seen throughout the hippocampal structures, being strongest in CA3 (Figure 6H,I), while hippocampal expression of *Opcml 1b* forms homogeneously dispersed puncta in CA and DG (Figure 6K,L).

In the adult brain, *Negr1*-expressing cells are observable throughout the sensory, motor, and visual cortices, while excluding the first layer (Figure 6M–O). The strongest signal is detectable in borders of the II/III layer, particularly in the barrel and auditory cortex (Figure 6M–O). Expression is observable in CA and DG of hippocampus exceptionally dominating in CA2 and DG regions (Figure 6N–R). The expression of *Ntm 1b*, *Opcml 1b*, and *Negr1* is maintained at layer 6b.

#### 2.3.2. Subcortical Structures and Diencephalon

Caudal sections of *Ntm 1a/1b* and *Opcml 1a/1b* (Figure 6C,F,I,L) illustrate mamillary hypothalamic regions with a different contour. Transcripts of both *Ntm 1a* and *1b* are ubiquitously present in most of the brain structures. Expression of *Ntm 1a* and *Ntm 1b* is diffusely localized to cortical and extended nuclei of amygdala (Figure 6B,C,E,F). In the rostral part of CPu *Ntm 1a* keeps dorso-ventral (Figure 6A) and *Ntm 1b* medio-lateral (Figure 6D) descending gradients.

Expression of *Ntm 1a* and 1b specifically avoids claustrum. Although *Ntm 1b* expression in thalamus and hypothalamus is seen as scattered puncta (Figure 6E,F), the expression of *Ntm 1a* diffuses from the centrolateral, centromedial, paracentral, rhomboid, reuniens thalamic nuclei to the subincertal nucleus (Figure 6B). Diffuse expression of *Ntm 1a* is also observable in the dorsal part of anterior pretectal nucleus and covers both the dorsal and ventral zona incerta (Figure 6C). Strong *Ntm 1a* expression is seen in the mamillary hypothalamic region (Figure 6C). Distinct expressions of *Ntm 1a* and *1b* are seen in habenula and paraventricular thalamic nuclei (Figure 6B,E).

Transcripts of *Opcml 1a* and *1b* are absent from the CPu and are expressed only weakly in regions of thalamus and hypothalamus. The mamillary hypothalamic region and habenula are positive for *Opcml 1a* (Figure 6H). In the amygdala, *Opcml 1a* is prominently expressed in basolateral nuclei, with weaker expression in the cortical amygdala (Figure 6H,I). *Opcml 1b* is absent in basolateral amygdala and only weakly expressed in the cortical amygdala (Figure 6K,L).

*Negr1* expression is absent or weak in CPu and strongly where the CPu transitions to the extended amygdala (Figure 6M,N). Distinct expression is seen in the hypothalamic nuclei (Figure 6M). Expression of *Negr1* covers the entire amygdala, being weakest in the basolateral nucleus (Figure 6N). Moderate expression is detectable in habenula and around mammillary recess of the third ventricle (Figure 6N).

#### 2.3.3. Cerebellum

The complementary expression pattern of IgLON members is clear in adult cerebellum (Figure 7). The expression of the *Lsamp 1a* isoform is not detectable (Figure 7A). However, *Lsamp 1b* expression is seen with X-Gal staining in the proximal segment of the molecular layer and invades through distinct cells in the Purkinje- and granular cell layer into the white matter (Figure 7B). Strong expression of *Ntm 1a* clearly delineates the Purkinje cell layer and is observable in specific cells of granular cell layer (Figure 7D). An intense *Ntm 1b* signal is evident only in granular cell layers (Figure 7E). The localization of *Opcml 1a* and *1b* transcripts occupies Purkinje and granular cell layers (Figure 7G,H,J). Weak labeling with *Opcml 1a* probe is evident in the molecular layer (Figure 7G). The *Negr1* signal is pronounced in the Purkinje cell layer and granular cell layer (Figure 7K). Immunostaining with antibodies raised against Lsamp, Ntm, and Opcml shows punctuated localization in the Purkinje and molecular cell layer (Figure 7C,F,I). Immunohistochemical staining of Negr1 shows strong staining of Purkinje cells and punctuated staining throughout the cerebellum (Figure 7L).

## 3. Discussion

The five members of the IgLON family of neuronal cell adhesion molecules carry molecular cues that guide axon and dendrite development, support the formation of synaptic connections, and promote the refinement of connectivity and plasticity during brain development. Use of multiple mRNA isoforms through alternative promoter usage is prevalent during embryonic brain development. mRNA isoforms encode a variety of proteins with diverse cellular and subcellular distribution over development stages, contributing significantly to a wide range of neuronal functions [4,5,43]. These isoform expression patterns appear to be important for normal brain development, since dysregulation of mRNA isoform expression is implicated in neurological and neuropsychiatric disorders [44,45]. Transcriptome-wide association studies suggest that isoform-level alterations show wider pathological effects as compared to gene-level alterations, enabling more precise analyses of brain function and development than those using solely gene-level analysis [43,45,46].

Our work highlights the selective usage of alternative IgLON promoters in different developmental stages and in various regions of the central nervous system. We found that two alternative isoforms-*1a* and *1b* in *Lsamp*, *Ntm* and *Opcml* are differentially expressed during development, indicating the regulation of alternative promoter usage. Previous neuroanatomical mRNA expression studies detected the origins of the diverse spatial distribution of IgLONs around E12.5–15, a time when neurons establish their circuits during embryonic brain development [12,41,47,48,49].Our data are the first to show the expression of IgLONs as early as E10.5. The earliest expression is detected from *Negr1* and from one of the alternative isoforms of *Lsamp*, *Ntm* and *Opcml* in the neuroepithelium (Appendix A). This timing correlates with the neurogenic and expansion phase of brain development. Soon afterward, complementary expression of alternative IgLON gene isoforms is seen in the cellular region of newly formed Cajal–Retzius cells (the first-born neurons from NSCs, formed after the expansion phase), suggesting that distinct IgLON isoforms play specific roles in the establishment of cortical architecture. The gradual increase and expansion of IgLONs throughout the developing brain during embryogenesis in the rodents (E13.5–E17) demonstrate the regionally dependent and complex cooperational usage of alternative promoters at the time when rapid neurogenesis is in progress and specific functional brain regions are being delineated. Expression patterns of IgLONs indicate the involvement during final stages of neurogenesis, when some of the differentiating daughter cells from NSCs undergo the gliogenesis phase of development and generate astrocytes, oligodendrocytes, and ependymal cells. Our result from whole-mount embryo also showed that some promoter-specific IgLONs expression continues laterally past the CNS and may represent migrating neural crest cells. Indeed, some of the earliest observations of IgLONs had been on migrating chick neural crest cells [50] as well as on chick dorsal root ganglia and sympathetic neurons, two trunk neural crest derivatives [51].

We have shown that IgLON molecules may be responsible for providing the adhesion differences that mediate the separation of the dorsal and ventral pallium compartments of the embryonic telencephalon. The telencephalon of amniotes is genetically divided into four evolutionarily conserved domains, identified as the medial, dorsal, lateral and ventral pallium [52] and we have shown that the expression of different IgLON isoforms is established in each of these four pallial subdivisions during early embryonic stages. Moreover, the expression of different IgLON isoforms in the germinal zone of dorsal pallium appears to participate in the development of the cortical layers. The IgLON isoforms expressions are clear in the ventricular zone of the lateral and ventral pallium subdivisions, subventricular zone and extend to developing cortices.

Above the subventricular zone, the crescent-shaped expression of *Lsamp uni* and *Ntm 1a/1b* probes appears at E13.5 in the lateral and ventral pallium, marking the boundary of the putative subplate. These expression domains are opposed with the expression of the *Negr1* domain, which lines the pial side. In addition to the subplate, cells derived from the lateral pallium contribute to development of the 6b layer of cerebral cortex, claustrum, and insular cortex, whereas the bed nucleus of stria terminalis, endopiriform nucleus, piriform cortices, and pallial amygdala are derived from ventral pallium. We observe both complementary and overlapping expression domains of IgLON alternative promoters in the primordia of aforementioned structures at E13.5. Importantly, these expression domains persist throughout the formation of these structures, and they are maintained in the adult brain. This gradient could be formed by several cell types such as the radial glial cells or migrating neurons. Determining the identity of these cells is crucial for understanding how brain regions arise during early neural development, as a similar gradient pattern of IgLON isoforms expression is observable along the pallial subdomains, in the insular/perirhinal mesocortex, the three layered allocortex, and the isocortex. This graded pattern of expression suggests an evolutionarily conserved mechanism, whereby regional coherence is achieved through the alternative and quantitative expression of cell adhesion molecules [53,54].

Differences in the isoform-specific expression of IgLONs were evident in the formation of the histogenetic fields of the hippocampus as it develops from the medial pallium. This region becomes critical for learning, memory formation, association, and cognition, as well as for sensorimotor processing, which will become executed in the isocortex (dorsal pallium in non-mammals). However, the lateral and ventral pallium derivatives, the claustrum, piriform cortex and the pallial amygdala, are involved in olfactory processing and control of emotional and autonomic behavior [55,56]. The observed developmental expression pattern of IgLONs in these brain substrates corresponds with eventual functional specificities, such as the regulation of social, emotional, and cognitive behaviors [20,22,34,35,36,37,38,39,40,41,42].

The development and maintenance of precise and specific neuronal contacts are critical for the proper functioning of neuronal cells [57], where IgLONs adhesion molecules exhibit key functions providing a platform for these cell–cell interacting processes. IgLONs are known to be involved in directing the formation of a variety of neural circuits. For example, Lsamp is involved in the fasciculation of dopaminergic afferents from the midbrain to the lateral habenula, thalamic, septo, and intrahippocampal circuits of limbic system circuitry. Ntm is involved in thalamocortical and pontocerebellar projections, while Negr1 is seen to be important in cortico-hippocampal circuits [12,18,21,22,34,42,58,59].

Regardless of individual expression patterns, the common peak temporal expression during early development suggests that the IgLON gene family plays important roles in many of the major neuro-developmental processes, including neuronal migration, neurite initiation, neurite outgrowth, axon pathfinding, axon fasciculation, synaptogenesis, myelination, cytoskeletal dynamics, and the maintenance of neuronal networks. Their distinct and partially overlapping spatial expression indicates that disruptions to the expression or function of these isoforms could produce additive deleterious effects such as abnormal neuritogenesis and neurogenesis, imbalanced excitatory/inhibitory ratios, and changes in brain volume. These changes can produce the diverse emotional, cognitive and social behavior disorders that are observed in the IgLON/sdeficient mice [20,21,22,34,35,36,37,38,39,40,41,42] used to model schizophrenia, autism, learning, and intellectual disabilities. However, the mechanisms behind these neuropathology are still unknown and needed further investigation.

Negr1 immunolabelling in the cerebellum is localized to the Purkinje cells and clearly reveals the arborization of their dendrites. Lsamp protein was localized as continuous puncta in the neurites reaching glia limitans. Similar localization patterns are seen in the glia limitans with Ntm and Opcml immunolabelling. These observations suggest a possible role for these IgLONs in the formation and maintenance of the blood-brain-barrier. Subsequent work is necessary to determine whether functional loss of IgLONs leads to defective glia limitans. This is particularly relevant as IgLONs have been found to have tumor-suppressive properties in neural and non-neural tissue [60,61,62,63,64,65,66]. Furthermore, NEGR1 has been found to control endothelial integrity in human brain microvessels [67]; LSAMP has been shown to be implicated in the coronary artery disease and both LSAMP and OPCML have been shown to be implicated in the epithelial-mesenchymal transition [68,69]. Taken together, current developmental data are an important addition to the earlier evidence suggesting that IgLONs are implicated in the evolutionary changes in the brain anatomy towards complexity including blood-brain-barrier permeability [70].

This paper allows several new questions to be analyzed, as further study is required to identify the isoform-specific interacting partners of IgLONs in the embryonic brain, the particular cell types expressing these isoforms, the mechanisms for the promoter-specific transcriptional regulation, and the molecular mechanisms of neural pathogenesis at the isoform-, rather than gene-level, resolution.

## 4. Materials and Methods

### 4.1. Animals

Wild-type C57BL/6 (Scanbur, Karl-slunde, Denmark) mice were used for in situ hybridization. Immunohistochemistry was performed on male mice in F2 background [(129S6/SvEvTac × C57BL/6) × (129S6/SvEvTac × C57BL/6)]. Mice were housed under standard laboratory conditions of a 12 hr light/dark cycle with lights on at 7:00 a.m., and they had ad libitum access to food (R70, Lactamin AB, Vadstena, Sweden) and water. Mice were mated and the presence of a vaginal plug was considered to be embryonic (E) day 0.5. Breeding and housing of the mice was conducted at the animal facility of the Institute of Biomedicine and Translational Medicine, University of Tartu, Tartu, Estonia. All animal procedures were performed in accordance with the European Communities Directive (86/609/EEC) and permit (No.29, 28 April 2014) from the Estonian National Board of Animal Experiments.

### 4.2. In situ Hybridization

The *Lsamp 1a* and *universal (uni*) probes were prepared as described in [41]. Mouse cDNA fragment specific for *Ntm 1a* (285 bp), *Ntm 1b* (500bp), *Opcml 1a* (492 bp), *Opcml 1b* (514 bp) and *Negr1* (650 bp) transcripts were cloned from a cDNA pool of C57BL/6 mouse brain and inserted into pBluescript KS+ vector (Stratagene, La Jolla, CA). We used the following primers (containing restriction sites):
Ntm1a For TATAGCGGCCGCGAGTATGAGTGGAGATAATTACGGANtm1a Rev TATAGTCGACCTTGGAAGAGGCACAGAGCCNtm1b For TATAGCGGCCGCGCTGGATTCAACCCAGCCACNtm1b Rev TATAGTCGACGTGGGTACAAGGAATAGCAGCCOpcml1a For TATAGCGGCCGCGGTGTGCCCATGCGAAGCACOpcml1a Rev TATAGTCGACGGATGAAGAGCAGGGCAGTGOpcml1b For TATAGCGGCCGCTCCTTTCTGTCAGAGACACTTGCOpcml1b Rev TATAGTCGACTGGGTACAAGGAATAGCAGCCTGNegr1 For TATAGCGGCCGCATGGTGCTCCTGGCGCAGGNegr1 Rev TATAGTCGACCAGCCTGGTCCCTTGTAATTCCAT

The mouse brains were dissected immediately after decapitation and fixed for 4 days in cold 4% PFA (Acros Organics, Ward Hill, MA, U.S.A)/PBS (pH 7.4). In newborn and adult mice, one temporal lobe was removed to allow better access to the fixative. The mouse brains were cryoprotected for 2 days in 30% sucrose (AppliChem, Dresden, Germany) in 4% PFA/PBS, and stored at −80 °C until sectioning. The non-radioactive in situ hybridization on 40-µm free-floating mouse (E17, P0, adult) brain cryo-sections using digoxigenin-UTP (Roche, Indianapolis, IN, U.S.A)-labeled RNA probes was performed as described in [71].

For the whole-mount in situ hybridization, the embryos were dissected in cold PBS and fixed in cold 4% PFA/PBS (pH 7.4) for 4 days. Fixed embryos were washed with PBS containing 0.25%, Tween 20 (Naxo, Tartu, Estonia) (PBST), followed by dehydrated in 25%, 50%, 75% methanol (Naxo, Tartu, Estonia) in PBST, and 100% methanol for 5 min at each step. Thereafter, the samples were rehydrated with methanol/PBST in reverse series, PBST washing and treatment with 10 µg/mL of proteinase K (AppliChem, Dresden, Germany) in PBST for 10 min is followed by washing with 2mg/mL of glycine (Naxo, Tartu, Estonia) in PBST. Embryos were then refixed in 4% PFA/0.2% glutaraldehyde (AppliChem, Dresden, Germany) for 20 min. After washing with PBST embryos were incubated in a prehybridization solution containing 50% formamide (AppliChem, Dresden, Germany)/5X SSC (pH 5) overnight at 65 °C with gentle agitation.

For hybridization RNA probe (500ng/mL) was added to the prehybridization solution and incubated over 3 days with gentle agitation at 65 °C. Samples were washed as follows: thrice with 1% SDS/5X SSC/50% formamide for 30 min at 65 °C; thrice with 2X SSC/50% formamide for 30 min and then overnight at 65 °C; thrice with TBST (0.15M NaCl: 0.1M Tris-HCl, pH7.5; 0.1% Tween-20) for 10 min each and incubated two days at 4 °C in anti-DIG-AP-conjugated antibody (1:1000, Roche, Indianapolis, IN, USA)/TBST. Unbound antibody was removed by three 10 min and five 60 min washes in TBST. Embryos were washed thrice with NTMT (0.1M NaCl; 0.1M Tris-HCl, pH9.5; 50 mM MgCl2; 0.1% Tween-20) for 10 min followed by incubation in the staining solution (BM-Purple, Roche, Indianapolis, IN, USA).

For cutting 50-µm vibratome sections, the stained (E10.5, 13.5) embryos were inserted into 1 mL of 0.5 % gelatine Sigma Aldrich, Taufkirchen, Germany/30 % BSA Sigma Aldrich, Taufkirchen, Germany/20 % sucrose (AppliChem, Dresden, Germany)/PBS, wherein 140 µL of 25 % glutaraldehyde (AppliChem, Dresden, Germany) was added immediately before insertion and incubated for 10 min. The sections were mounted into 70 % glycerol and micro photographed using an Olympus BX61 microscope equipped with an Olympus DX70 CCD camera (Olympus, Hamburg, Germany).

### 4.3. X-Gal Staining

Free-floating sections were stained overnight for X-Gal staining for detecting the distribution of *Lsamp 1b* promoter was performed as described previously [72]. Alternatively, whole brains were incubated in X-Gal staining solution immediately after fixation. After X-Gal staining, tissue was incubated in 2% PFA solution in PB to give it a pale white appearance. Sections were transferred to gelatinized glass slides and mounted with Pertex (Histolab, Malmö, Sweden)

### 4.4. Immunohistochemistry

Brains of adult mice were dissected and fixed in 4% PFA for 4–5 days followed by cryoprotection with 30% sucrose in PBS, frozenand sectioned at 40 µm. Free-floating coronal sections from the cerebellar paraflocculus region were permeabilized, blocked and immunostained with primary antibodies: mouse anti-Lsamp (1:200, DSHB; 2G9), goat anti-Opcml (1:200, Santa Cruz Biotechnology, Heidelberg, Garmany; sc-26121; this product has been discontinued), mouse anti-Negr1 (1:100, Santa Cruz Biotechnology, Heidelberg, Garmany; sc-393293), mouse anti-Ntm (1:200, Santa Cruz Biotechnology, Heidelberg, Garmany; sc390941) followed by secondary antibodies: Rhodamine (TRITC)-AffiniPure Donkey Anti-Mouse IgG (H+L) (1:500, Jackson ImmunoResearch Labs, West Grove, PA, USA; 715-025-150) and Rhodamine Red-X-AffiniPure Fab Fragment Rabbit Anti-Goat IgG (H+L) antibody (1:500, Jackson ImmunoResearch Labs, West Grove, PA, USA; 305-297-003). Nuclei were stained with 5 μg/mL Bisbenzimide H 33258 (Hoechst 33258, Sigma Aldrich, Taufkirchen, Germany)/PBS for 15 min. Subsequently, sections were washed with PBS and mounted with Fluoromount mounting medium (Sigma Aldrich, Taufkirchen, Germany), and covered with a 0.17-mm coverslip (Deltalab, Barcelona, Spain). Specificity of the immunohistochemistry was determined by incubations without the primary antibodies. Fluorescent images were obtained with the Olympus FV1200MPE (Olympus, Hamburg, Germany) laser scanning confocal microscope. Abbreviations in all the figures representing anatomical data have been adopted from the mouse brain atlas [73].

## 5. Conclusions

In conclusion, we have shown that IgLON members are involved in, and perhaps orchestrate, the early events of pallium development. The complexity of the spatiotemporal distribution patterns of the IgLONs is intimately linked to regional brain organization and to the emergence of functional distinctions in the developing nervous system. Distinct isoform-specific expression domains and gradients suggest the importance of IgLONs alternative promoter usage in helping coordinate the complexly integrated functions of neural cell proliferation, differentiation, and morphogenesis into the more specialized substructures.Importantly, IgLON expression patterns (1) may be responsible for creating the anatomical and functional boundaries between regions (which may then be stabilized by other molecules), (2) may help stabilize pre-existing boundaries between anatomical and functional regions, or (3) may be a consequence of the anatomical and functional regions, perhaps acting as tumor suppressing inhibitors of cell proliferation and apoptosis. Indeed, it would not be surprising if IgLONs were performing different roles in different brain regions. The IgLON superfamily may have evolved the alternative promoter system to produce necessary mRNA isoforms from single gene loci in the specific histogenetic fields at specific times.

## Figures and Tables

**Figure 1 ijms-22-06955-f001:**
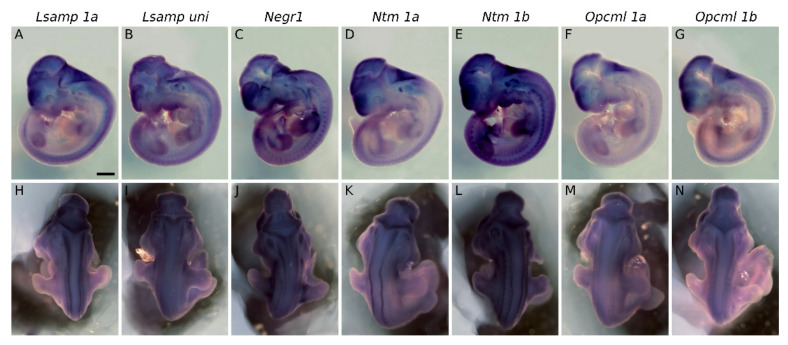
Expression of IgLON family alternative promoters at E11.5. Expression of *Lsamp 1a*, *Lsampuni*, *Negr1*, *Ntm 1a*, *Ntm 1b*, *Opcml 1a*, and *Opcml 1b* mRNA was detected by whole-mount in situ hybridization of wild-type embryos. (**A**–**G**) lateral and (**H**–**N**) dorsal views of whole-embryos. Observed mRNA expression is seen bilaterally (**H**,**I**,**K**,**L**) along neural tube or (**J**,**M**,**N**) at the level of limbs. All observed mRNA probes are expressed in forebrain region and pharyngeal arches. (**A**–**E**) specific expression of corresponding probes is seen in the developing inner ear. (**A**,**B**,**E**,**G**). Faint expression is seen (**A**,**B**,**E**,**G**) inside and around the developing eye. Scale bar: 1 mm.

**Figure 2 ijms-22-06955-f002:**
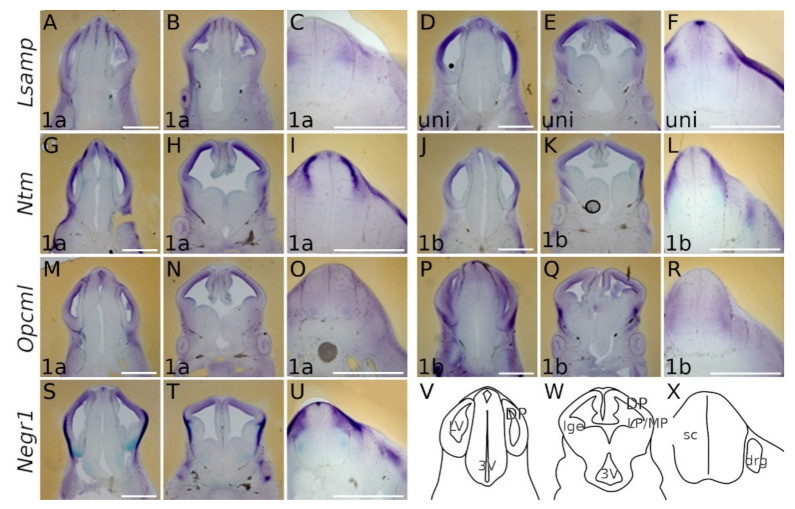
Expression from IgLON alternative promoters in E13.5 mouse CNS. Whole-mount in situ mRNA hybridization. (**A**,**D**,**G**,**J**,**M**,**P**,**S**) coronal sections through caudal telencephalon and (**B**,**E**,**H**,**K**,**N**,**Q**,**T**) telencephalon at intermediate level through interventricular foramen (laterally), showing preoptic area, hypothalamus and diencephalon (medial portions). All observed IgLON transcripts are present in DP from ventricular to pial surface along the rostral-caudal axis; *Negr1* and *Opcml 1b* give strong signal also in LP. (**C**,**F**) *Lsamp 1a/uni* expression is seen dorso-laterally in developing sensory input area of sc while *Lsampuni* marks the roof plate. (**I**,**L**) *Ntm 1a* and *1b* occupy the dorsal portion of the neural tube rather complementary: *Ntm 1a* is strongly expressed below the *Ntm 1b* domain. (**O**) weak uniform *Opcml 1a* signal is detectable through the developing sc, being somewhat stronger in the motor region (**R**) *Opcml 1b* occupies the lateral portion of sc. (**U**) *Negr1* expression is complementary to (**C**,**F**) *Lsamp 1a/uni* with the exception seen at roof plate. (**V**–**X**) schemes of cutting planes. Abbreviations: sc, spinal cord; DP, dorsal pallium; LP, lateral pallium; VP, ventral pallium; lge, lateral ganglionic eminence; *Lsampuni (Lsamp 1a + 1b)*; LV, lateral ventricle; 3V, third ventricle; drg, dorsal root ganglion. Scale bar: 1 mm.

**Figure 3 ijms-22-06955-f003:**
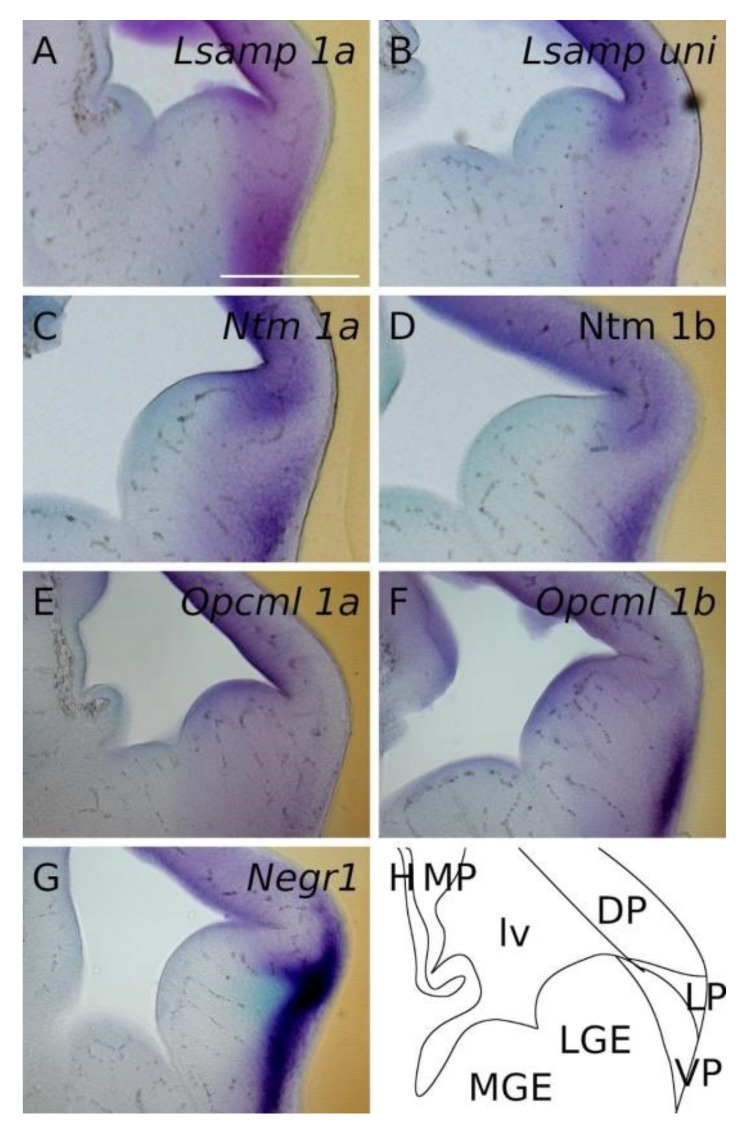
In situ mRNA hybridization displaying IgLON members in developing pallium. (**A**–**G**) Expression of IgLON probes recapitulating alternative promoter activity in pallial divisions at E13.5 mouse embryo. (**H**) scheme of cutting planes. Abbreviations: DP, dorsal pallium; LP, lateral pallium; VP, ventral pallium; MP, medial pallium; lv, lateral ventricle; LGE, lateral ganglionic eminence; MGE, medial ganglionic eminence; *Lsampuni, Lsamp 1a + 1b*. Scale bar: 500 μm.

**Figure 4 ijms-22-06955-f004:**
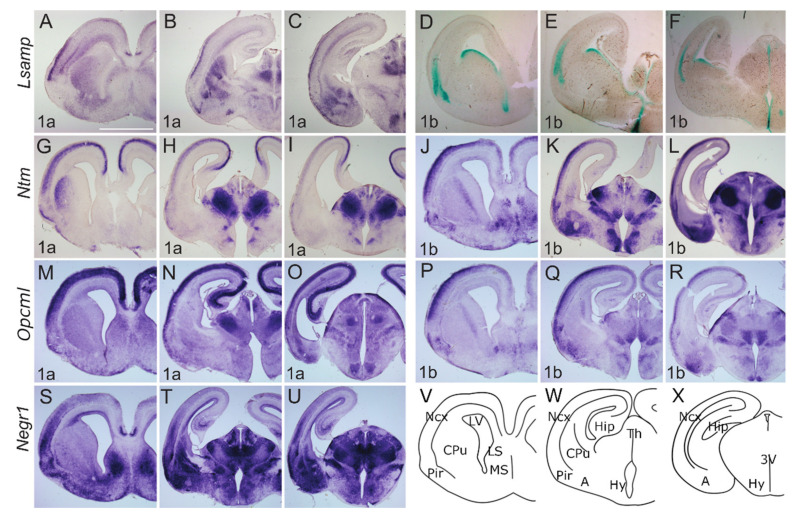
Expression from IgLON alternative promoters at E17 mouse CNS. (**A**–**C**;**G**–**U**) In situ mRNA hybridization; (**D–F**) X-gal staining. (**A**,**D**,**G**,**J**,**M**,**P**,**S**) coronal sections through the level of olfactory tuberculum and septum. (**B**,**E**,**H**,**K**,**N**,**Q**,**T**) sections pass through the middle part of Amy and diencephalon; (**C**,**F**,**I**,**L**,**O**,**R**,**U**) through the caudal part of Amy and diencephalon. *Lsamp 1a* is strongly expressed laterally in upper layers of neo- and allocortex as well as in subplate region; the expression is seen in developing CPu, Amy; and thalamic nuclei. (**D**–**F**) Overall the expressions of *Lsamp 1a* and *1b* are largely complementary: *Lsamp 1b* expression is visible at the ventral portion of lateral ventricles and moving caudally, lines the third ventricle. (**G**–**I**) *Ntm 1a* and (**J**–**L**) *Ntm 1b* are strongly detected at upper layers of the developing cortex. (**G**) in CPu, *Ntm 1a* is seen laterally beneath the developing white matter, whereas (**J**) *Ntm 1b* is seen in the more internal part of CPu. Strongest expression of (**H**,**I**) *Ntm 1a* and (**K**,**L**) *Ntm 1b* is seen in thalamic nuclei. Unlike *Ntm 1a* (**K**,**L**) the expression of *Ntm 1b* is observable in Amy. (**M**–**O**) *Opcml 1a* is broadly expressed in developing cortex and hippocampus; however (**P**–**R**) *Opcml 1b* is restricted to dorsolateral cortex with gradient starting from developing 2/3 layer. (**N**,**O**) *Opcml 1a* expression in thalamic and hypothalamic nuclei is complementary to (**Q**,**R**) *Opcml 1b*. (**S**–**U**) *Negr1* is broadly expressed in the lateral region of the developing cortex, excluding the uppermost layer, and in the dorsal part of LV ventricular zone. (**T**,**U**) strong expression is seen throughout Amy and Th. (**V**–**X**) schemes of cutting planes. Abbreviations: 3V, third ventricle; Amy, amygdala; CPu, caudal putamen; LV, lateral ventricle; MS, medial septum; Ncx, neocortex; Pir, piriform cortex; Th, thalamus; Hy, hypothalamus. Scale bar: 1 mm.

**Figure 5 ijms-22-06955-f005:**
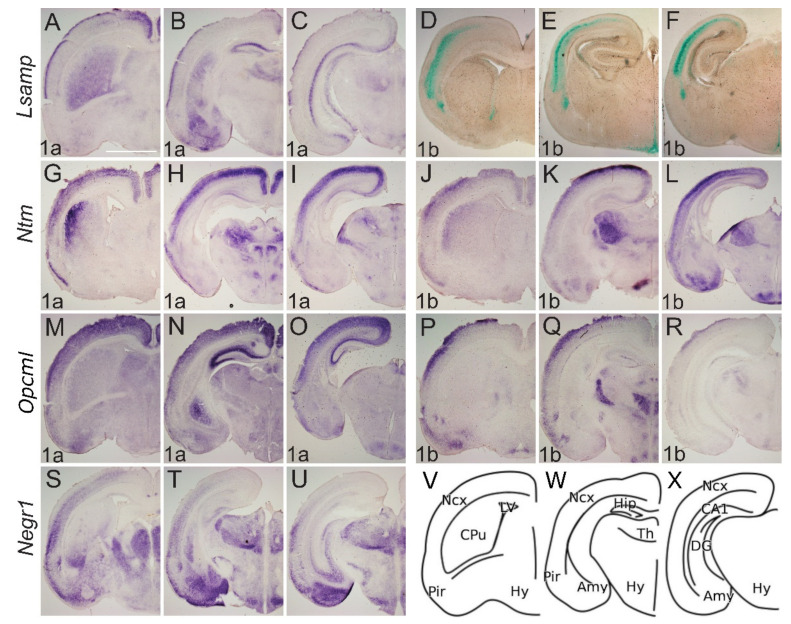
Expression from IgLON alternative promoters at P0 mouse CNS. (**A**–**C**; **G**–**U**) In situ mRNA hybridization; (**D**–**F**) X-gal staining. (**A**,**D**,**G**,**J**,**M**,**P**,**S**) coronal sections through the level of olfactory tuberculum and septum. (**B**,**E**,**H**,**K**,**N**,**Q**,**T**) sections pass through the middle part of Amy and diencephalon; (**C**,**F**,**I**,**L**,**O**,**R**,**U**) through the caudal part of Amy and diencephalon. (**A**–**C**) *Lsamp 1a* is expressed laterally in the upper layer of the cortex and in the ammon horns of the hippocampus. Expression in CPu forms a medio-lateral gradient where the strongest expression is seen medially. Strong expression is seen in Amy. (**D**–**F**) *Lsamp 1b* in the cortex is seen below *1a* expression domain and is missing from Amy. (**E**,**F**) expression of *Lsamp 1b* persists around the third ventricle. (**G**–**I**) *Ntm 1a* in cortical layers is broad and forms gradient being strongest at dorso-medial side. Strong expression is seen in Pir. Same pattern is seen with (**J**,**K**) *Ntm 1b*; however, the expression is more restricted to upper layers of cortex. (**G**,**J**) in striatum *Ntm 1a/1b* is seen laterally immediately beneath the developing corpus callosum. Complementary expression of (**H**,**I**) *Ntm 1a* and (**K**,**L**) *Ntm 1b* is revealed in thalamic nuclei and Amy. (**M**–**O**) *Opcml 1a* is broadly expressed in the cortex, being strongest in the hippocampus. (**P**–**R**) *Opcml 1b* is restricted to the dorsolateral surface of the cortex and portion of thalamic nuclei. (**S**–**U**) *Negr1* is broadly expressed in lateral surface of developing neo- and piriform cortex; (**U**) *Negr1* expression in hippocampus is observable at caudal portion and is (**T**,**U**) strong in Amy and upper thalamic nuclei. (**V**–**X**) schemes of cutting planes. Abbreviations: Amy, amygdala; CA1, cornuammonis region 1; DG, dentate gyrus; CPu, caudal putamen; LV, lateral ventricle; Ncx, neocortex; Pir, piriform cortex; Th, thalamus; Hy, hypothalamus. Scale bar: 1 mm.

**Figure 6 ijms-22-06955-f006:**
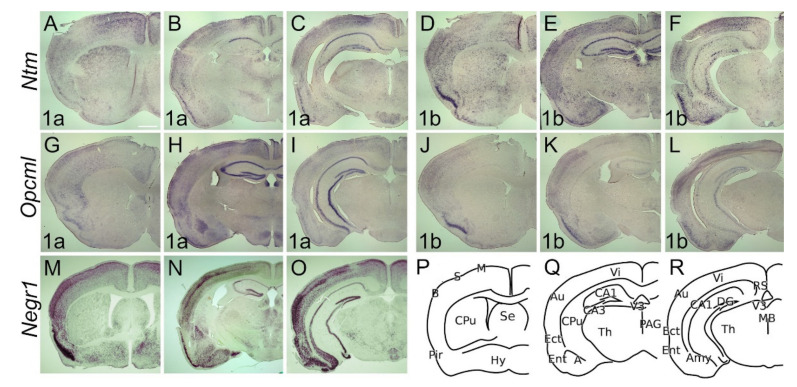
Expression from IgLON alternative promoters in the adult mouse CNS. In situ mRNA hybridization. (**A**,**D**,**G**,**J**,**M**) coronal sections through the level of olfactory tuberculum and septum. (**B**,**E**,**H**,**K**,**N**) sections through the middle part of Amy and diencephalon region. (**C**,**F**,**I**,**L**,**O**) caudal sections illustrate the mammillary hypothalamic region. (A) *Ntm 1a* expression is seen in Pir and anterior nuclei of thalamusas well as defined layers of sensory and motor cortex. (**B**,**C**) Observable *Ntm 1a* expression pattern is seen in visual and auditory cortex and ecto- and entorhinal cortices. Relatively strong expression is seen in amygdala as well as in specific cells of the hippocampus. (**D**,**E**) Similar expression pattern is seen with *Ntm 1b* probe, where strong expression is defined in the auditory cortex and amygdala. (**G**,**H**) *Opcml 1a* is broadly expressed in the cortex and strongest at upper layers throughout sensory and motor columns. Intense staining is seen in (**G**) piriform cortex and (**H**,**I**) hippocampus. Similarly (**J**–**L**) *Opcml 1b* is restricted to the upper layers of the sensory cortex. Strongest expression is seen in (J) piriform cortex. Faint expression is seen in (**K**,**L**) hippocampus. (**M**,**N**) *Negr1* is broadly expressed in sensory cortical structures, notably in layers of the auditory cortex. Intense expression is seen in (**M**) piriform- and entorhinal cortex. (**N**,**O**) Relatively strong signal is seen in the hippocampus, notably in CA2 and dentate gyrus. Strong expression in amygdala greatly overlaps with other family members. Faint expression is observable in thalamic nuclei. (**P**–**R**) schemes of cutting planes. Abbreviations: 3V, third ventricle; Amy, amygdala; Au, auditory cortex; B, barrel cortex; CA1, *cornuammonis* region 1; CA3, *cornuammonis* region 3; CPu, caudal putamen; DG, dentate gyrus; Ect, entorhinal cortex; Ent, entorhinal cortex; LV, lateral ventricle; MS, medial septum; Ncx, neocortex; Pir, piriform cortex; Th, thalamus; Hy, hypothalamus. Scale bar: 1 mm.

**Figure 7 ijms-22-06955-f007:**
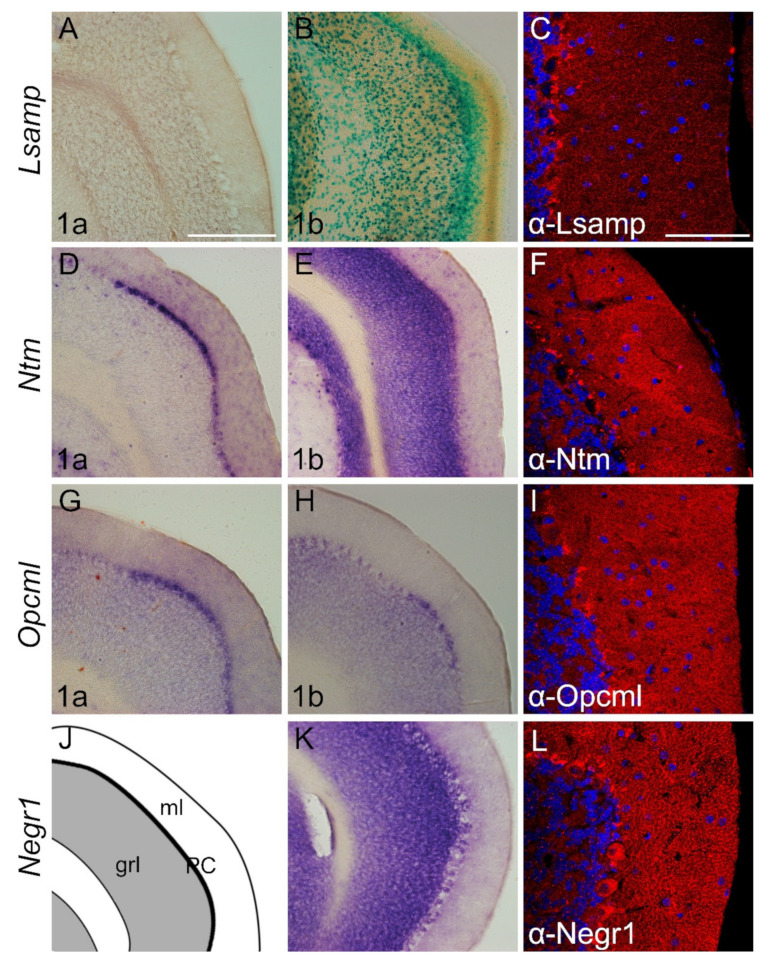
Expression from IgLON alternative promoters and proteins in adult mouse cerebellum. (**A**,**D**–**E**,**G**,**H**,**K**) in situ mRNA hybridization; (**B**) X-gal staining; (**C**,**F**,**I**,**L**) immunofluorescence staining. (**A**–**L**) Images present the expression in the paraflocculus region of the cerebellum. (**A**) *Lsamp 1a* expression is not observable; however (**B**) *Lsamp 1b* is highly expressed through molecular and granular cell layers. (**D**) *Ntm 1a* expression is restricted to the Purkinje cell layer whereas (**E**) *Ntm 1b* is observable in the granular cell layer. As with *Ntm 1a* (**G**,**H**) *Opcml 1a/1b* is expressed in defined locations of Purkinje cell layers. (**K**) *Negr1* probe is observable in both Purkinje cells and granular cell layers. (**J**,**K**)scheme of cutting planes. Abbreviations: grl, granular layer; ml, molecular layer; PC, Purkinje cell layer. Scale bar: 300 μm (**A**,**B**,**D**,**E**,**J**,**K**); 100 μm (**C**,**F**,**I**,**L**).

**Table 1 ijms-22-06955-t001:** Relative abundance of IgLON transcripts in developing cortex and hippocampus at E17 and P0. Expression abundance is determined after mRNA hybridization by visual observation as +—weak; ++—moderate; +++—strong; -—absent. Relative abundance is not comparable between different transcripts. *Lsamp 1b**—X-Gal staining. Abbreviations: CA, *cornuammonis*; CP, cortical plate; DG, dentate gyrus; IZ, intermediate zone, MZ, marginal zone; L, presumptive cortical layers; SVZ, subventricular zone; SP, subplate; VZ, ventricular zone.

Lateral Neocortex and Hippocampus at the Level of Olfactory Tuberculum and Septum															
	MZ (L1)		CP				SP (L6B)		IZ	SVZ	VZ	Hippocampus			
			OUTER (L2/3)		MIDDLE (L4—6)							CA		DG	
	E17	P0	E17	P0	E17	P0	E17	P0	E17	E17	E17	E17	P0	E17	P0
***Lsamp 1a***	-	+++	+++	+	+	+	++	++	+	+	-	+	+++	+	++
***Lsamp 1b*** *****	-	-	-	+++	++	++	-	-	-	-	-	-	-	-	-
***Ntm 1a***	-	+	++	+++	++	+++	+	+	-	-	-	+	+	-	-
***Ntm 1b***	+	+	+++	++	++	+	+	-	-	+	+	+	+	+	-
***Opcml 1a***	+++	+++	+++	+++	++	++	+	-	-	+	+	+++	+++	+	-
***Opcml 1b***	-	+++	+++	+++	++	+	+	++	-	+	+	+	+	+	++
***Negr1***	+	-	+++	++	+++	++	+++	+	-	+++	+	++	+	+	+

## Data Availability

Not applicable.

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
