# Peer review of "Alternative Promoter Use Governs the Expression of IgLON Cell Adhesion Molecules in Histogenetic Fields of the Embryonic Mouse Brain"

_ijms, 2021, doi:10.3390/ijms22136955_

Round 1

Reviewer 1 Report

This paper is represents the logic continuation of the work initiated by the same group on promoter-specific expression of IgLON family members (Front. Neurosci. 2017 11;38). The paper shows the expression patterns of different forms of members of neuronal cell adhesion molecules of the IgLON superfamily during mouse brain embryonic and postnatal development. The analysis is entirely performed by in-situ hybridization and sections are analysed on low magnification. As such the paper is very descriptive and does not provide informations particularly useful to understand the functions of the members of this family of genes.

The analysis is only limited to the central nervous system, however these genes are also expressed in  other tissues and it might be interesting to see, particularly in early stages of development ,the pattern of expression in the whole embryo.

Furthermore, I would expect, as a minimum, to see some double staining in order to identify which major cell populations are involved. For example, in figure 1 some of the genes seem to be expressed along the pathways of migratory neural crest cells (cranial and/or somatic). This could be a very important information, however at low magnification and without a double staining it is difficult to say. In the same line we do not know which neuronal or glial populations are involved.

In my view the discussion lacks of "functional" considerations. All of these genes have been inactivated in the mouse showing  anomalies due to defects of corticogenesis, neuritogenesis, axon guidance, myelinisation, as well as adult abnormalities of neurogenesis, behaviour and socialisation. These data should be considered much better in the discussion of this paper in order to give some interest to these observations.

Author Response

We wish to thank the reviewer for their thoughtful comments. Indeed they have asked questions that we have also asked in the manuscript and which we hope to address in future studies- namely the identity of the cell types bearing the different IgLON isoforms and whether the IgLONs staining cells lateral to the CNS represent a particular type of neural crest cell.

This manuscript has been checked by our native English-speaking coauthor.

Changes made in the manuscript are with track change.

Response to the Comments:

Point 1: The analysis is only limited to the central nervous system, however these genes are also expressed in other tissues and it might be interesting to see, particularly in early stages of development,the pattern of expression in the whole embryo.

Response 1: Indeed, according to the valuable notion by the reviewer, different IgLON isoforms are expressed also in the periphery during early development. This is a wider subject with somewhat different focus; we are in process of preparation of a separate paper about it.

Our earlier published article (Vanaveski et al. 2017) also describes promoter-specific expression of IgLONs in 12 tissues other than the brain, using a customized 5′-isoform-specific quantitative real-time PCR (qPCR) assay in adult mice. However, as per the suggestion by reviewer, we incorporated the expression pattern of IgLON isoforms in the early stage of embryonic development (Whole mount E11.5 embryos; Figure 1 in result section, p-3). Since our present study is focused on spatial distribution of IgLON isoforms in the embryonic brain, we have concentrated on presenting the results of distinct developmental stages of the mouse brain.

Point 2: Furthermore, I would expect, as a minimum, to see some double staining in order to identify which major cell populations are involved. For example, in figure 1 some of the genes seem to be expressed along the pathways of migratory neural crest cells (cranial and/or somatic). This could be a very important information, however at low magnification and without a double staining it is difficult to say. In the same line we do not know which neuronal or glial populations are involved.

Response 2: We have also noticed that some promoter-specific IgLON expression continues laterally past the central nervous system (e.g., Figure 1, 2) and may represent migrating neural crest cells. Indeed, some of the earliest observations of IgLONs had been on migrating chick neural crest cells (Kimura et al., 2001) as well as on chick dorsal root ganglia and sympathetic neurons, two trunk neural crest derivatives (Lodge et al, 2000).We have included this material in the discussion section of our revised manuscript (p-13).

The revised manuscript also contains newly added data from whole mount embryo staining (Figure 1, p-3) that similarly suggest that the expression of IgLON isoforms might be in the migrating neural crest cells. Our present paper, however, had to be specifically focused on the central nervous system. To include neural crest derivatives would have been a huge undertaking that would have had to involve multiple markers. Double staining would only provide a first approximation, since there are no specific neural crest cell markers, and even HNK-1 is found on other cell types. So we decided that we would confine our paper to the nascent divisions of the developing mouse brain. Resolving the fine anatomy of alternative promoter-specific 1a and 1b mRNA isoforms in LsampOpcmlNtm and the single promoter of Negr1 in the murine brain has produced a paper that is already long and detailed. So while the question of whether these stained cells lateral to the neural tube are neural crest derivatives is worth exploring, we decided it was (literally) peripheral to our research, and if done well, would expand the paper to an unpublishable length. As mentioned earlier, we will mention the possibility that these IgLON-expressing cells are derived from the neural crest and will cite these relevant papers.

According to the reviewer's suggestion, we included additional images of higher magnification (Supplementary Figure S2) showing closely the intermediate level of the brain to provide a better perspective for determining the cell types. Several brain regions show robust and abundant expression of separate IgLON isoforms, indicating that the IgLON members are not confined to specific neural or glial cell types. Establishing the presence of different transcripts according to cell types would require a separate in depth study and is a research question to be clarified in future studies.

Point 3: In my view the discussion lacks of "functional" considerations. All of these genes have been inactivated in the mouse showing  anomalies due to defects of corticogenesis, neuritogenesis, axon guidance, myelinisation, as well as adult abnormalities of neurogenesis, behaviour and socialisation. These data should be considered much better in the discussion of this paper in order to give some interest to these observations.

Response 3: As per the reviewer's suggestion, the discussion has been improved by taking functional considerations in account. Additional paragraphs added to the page 14 (discussion) of the manuscript are as mentioned below:

The development and maintenance of precise and specific neuronal contacts are critical for the proper functioning of neuronal cells [Lodato et al. 2014], where IgLONs adhesion molecules exhibit key functions providing a platform for these cell–cell interacting processes. IgLONs are known to be involved in directing the formation of a variety of neural circuits. For example, Lsamp is involved in the fasciculation of dopaminergic afferents from the midbrain to the lateral habenula, thalamic, septo- and intrahippocampal circuits of limbic system circuitry. Ntm is involved in thalamocortical and pontocerebellar projections, while Negr1 is seen to be important in cortico-hippocampal circuits (Keller et al., 1989; Pimenta et al., 1995; Mann et al., 1998, Schmidt et al., 2014, Struyk et al., 1995; Chen et al., 2001, Singh et al. 2018, Noh et al. 2019 Szczurkowska et al. 2018).

Regardless of individual expression patterns, the common peak temporal expression during early development suggests that the IgLON gene family plays important roles in many of the major neuro-developmental processes, including neuronal migration, neurite initiation, neurite outgrowth, axon pathfinding, axon fasciculation, synaptogenesis, myelination, cytoskeletal dynamics and the maintenance of neuronal networks. Their distinct and partially overlapping spatial expression indicates that disruptions to the expression or function of these isoforms could produce additive deleterious effects such as abnormal neuritogenesis and neurogenesis, imbalanced excitatory/inhibitory ratios, and changes in brain volume. These changes can produce the diverse emotional, cognitive and social behavior disorders that are observed in the IgLON/s deficient mice (Innos et al. 2012, 2013, Philips et al. 2015, 011, Mazitov et al. 2017, Singh et al. 2018, 2019, Bregin et al. 2020) used to model schizophrenia, autism, learning and intellectual disabilities However, the mechanisms behind these neuropathologies are still unknown and needed further investigation.

Reviewer 2 Report

The manuscript "Alternative promoter use governs the expression of IgLON cell adhesion molecules in histogenetic fields of the embryonic mouse brain", By Toomas Jagomäe et al., is a very interesting work that presents for the first time information about how IgLON cell adhesion molecules are regulated by alternative promoter usage in embryonic mouse brain. They also suggested relation between IgLON and development of neuropsychiatric disorders.

The work is very well written and presented, and I have just a minor comment:

Is there any mechanism of neural pathogenesis dependent with expression timing of IgLON?

Author Response

We wish to thank the reviewer for their thoughtful comments. Indeed they have asked questions that we have also asked in the manuscript and which we hope to address in future studies- namely the identity of the cell types bearing the different IgLON isoforms and whether the IgLONs staining cells lateral to the CNS represent a particular type of neural crest cell.

This manuscript has been checked by our native English-speaking coauthor.

Changes made in the manuscript are with track change.

Response to the Comments:

Point 1: Is there any mechanism of neural pathogenesis dependent with expression timing of IgLON?

Response 1: The mechanisms of neuronal pathogenesis pertaining to IgLONs are still to be elucidated. We get hints about the pathogenesis of IgLON deficiencies by doing experiments using knockout mice, where the anomalies caused by the absence of the gene are studied in the cellular, behavioral and physiological levels. However, the roadblock to studying the possible roles of IgLON expression timing is that there are no conditional knockout mice available for the IgLONS. We appreciate that reviewer have pointed out this important question which we are trying to address in our future research.

We included few additional sentences in the introduction section (p-2) related to human neuropathology associated with IgLONs as below:

In humans, several polymorphisms at IgLON loci and imbalances in IgLON expression levels are associated with cognitive ability and  wide variety of neuropsychiatric disorders, such as schizophrenia, major depression, bipolar disorder, and autism [23–30]. Specific phenotypes appear to be influenced by all IgLON genes. For example, autism has been shown to be linked with the loci of NEGR1 [31] NTM/OPCML [32] and LSAMP [33]. Similarily, differences in cognitive ability have been associated with NEGR1 [34] and the loci of all IgLONs significantly associate with the educational attainment [35].

Two additional paragraphs in the discussion section (p 14) of the manuscript in order to describe the precise role of IgLONs and the expression analysis in relation to the neuropathology of brain disorder, as below:

The development and maintenance of precise and specific neuronal contacts are critical for the proper functioning of neuronal cells [Lodato et al. 2014], where IgLONs adhesion molecules exhibit key functions providing a platform for these cell–cell interacting processes. IgLONs are known to be involved in directing the formation of a variety of neural circuits. For example, Lsamp is involved in the fasciculation of dopaminergic afferents from the midbrain to the lateral habenula, thalamic, septo- and intrahippocampal circuits of limbic system circuitry. Ntm is involved in thalamocortical and pontocerebellar projections, while Negr1 is seen to be important in cortico-hippocampal circuits (Keller et al., 1989; Pimenta et al., 1995; Mann et al., 1998, Schmidt et al., 2014, Struyk et al., 1995; Chen et al., 2001, Singh et al. 2018, Noh et al. 2019 Szczurkowska et al. 2018).

Regardless of individual expression patterns, the common peak temporal expression during early development suggests that the IgLON gene family plays important roles in many of the major neuro-developmental processes, including neuronal migration, neurite initiation, neurite outgrowth, axon pathfinding, axon fasciculation, synaptogenesis, myelination, cytoskeletal dynamics and the maintenance of neuronal networks. Their distinct and partially overlapping spatial expression indicates that disruptions to the expression or function of these isoforms could produce additive deleterious effects such as abnormal neuritogenesis and neurogenesis, imbalanced excitatory/inhibitory ratios, and changes in brain volume. These changes can produce the diverse emotional, cognitive and social behavior disorders that are observed in the IgLON/s deficient mice (Innos et al. 2012, 2013; Philips et al. 2015; Mazitov et al. 2017; Singh et al. 2018, 2019; Bregin et al. 2020) used to model schizophrenia, autism, learning and intellectual disabilities However, the mechanisms behind these neuropathologies are still unknown and needed further investigation.

Round 2

Reviewer 1 Report

The authors have responded well to the comments, the paper could still be improved as it remains very descriptive. Still, I recommend publication as such as these data could be the basis for future studies.